# OpenReview forum: "BEAT: Visual Backdoor Attacks on VLM-based Embodied Agents via Contrastive Trigger Learning"
_ICLR.cc/2026/Conference — ICLR 2026 Poster_

### Official Review · Reviewer_24ke · 2025-10-28

**Soundness:** 3
**Presentation:** 3
**Contribution:** 2
**Rating:** 6
**Confidence:** 4

**Summary:**

The paper proposes BEAT, the framework to backdoor MLLM embodied decision-making systems (e.g., home robots). The paper proposes contrastive trigger learning (CTL) to tighten the trigger boundary. Then, it utilizes two-stage fine-tuning (SFT and CTL) to improve the benign performance on embodied tasks and inject the backdoor behaviors while ensuring a low false trigger rate. In the experiments, three models (i.e., GPT-4o, Qwen2 7B, and InternVL3-8B) are tested on two datasets (i.e., VAB-OmniGibson and EB-ALFRED), showing high benign performance, attack success rate, and low false trigger rate.

**Strengths:**

- **Interesting topic**: The paper addresses an interesting and critical topic of embodied agent backdoor attacks. It shows that MLLMs can be backdoored to perform dangerous tasks when a visual trigger is presented.
- **Good writing quality**: The paper clearly introduces the design challenge, the threat model, and the methodology. The scientific contribution is clear, and the content is easy to follow.
- **End-to-end evaluation**: The paper conducts an end-to-end evaluation on a simulator to demonstrate the effectiveness of the backdoor attack, where the agents perform malicious tasks when being visually triggered.

**Weaknesses:**

- **Lack of a comprehensive attack robustness discussion**: The paper discussed trigger robustness under an unseen trigger context. However, other factors influence the trigger robustness. For example:
  - *System prompt robustness*: The attacker can only specify finite system prompts, but the end-user can change to whatever they want to maximize the benign performance.
  - *Environment robustness*: The attacker can only train the backdoor models using finite environments, while the end user might deploy the model in different scenarios (e.g., different simulators or even a physical environment).
- **Lack of defense discussion**: The paper proposes the attack without discussing any defenses. Therefore, it’s unknown if the attack can be simply defended, which might limit the real-world usability. For example, what if the backdoor model is further fine-tuned by the end-user on their specific tasks [r1]?

---
**Reference**

[r1] Ni, Zhenyang, et al. "Physical backdoor attack can jeopardize driving with vision-large-language models." arXiv preprint arXiv:2404.12916 (2024).

**Questions:**

- **Need for clarification**: The author mentioned that a bounding box annotation is visualized in the image to help the MLLMs' planning process. However, this raises the doubt whether the model is truly triggered by the objects or it is triggered by the text. If the actual trigger is the textual annotation, it largely weakens the main contribution of the paper (i.e., visual backdoor attacks). Therefore, this part needs to be analyzed and put in the main paper for justification.

**Details Of Ethics Concerns:**

The author already included the ethics discussion.

---

> ### Author Response · Authors · 2025-11-21
> **Response (1/2)**
>
> Thank you for recognizing the importance of our research problem, the clarity of our writing, and the comprehensiveness of our experiments. We appreciate your valuable comments. Our responses are as follows:
>
> > Weakness 1: Lack of a comprehensive attack robustness discussion, including **system prompt robustness**: the attacker can only specify finite system prompts, but the end-user can change to whatever they want to maximize the benign performance; and **environment robustness**: The attacker can only train the backdoor models using finite environments, while the end user might deploy the model in different scenarios (e.g., different simulators or even a physical environment).
>
> Thank you for your insightful comments. We would first like to clarify the intended usage scenario of our finetuned models. In our threat model, the user chooses a specific model because it performs well on certain tasks under a particular task-specific finetuning. However, current MLLM-based agents exhibit limited transferability across environments: changing the environment (e.g., from VAB to EB-Alfred, or to the real world) can completely alter the underlying system problem, action space, and visual distribution. As a result, **cross-environment deployment is uncommon in practice and falls outside the scope of our threat model**.
>
> To evaluate system prompt robustness **within the same environment**, we asked GPT to “Rewrite this prompt for an MLLM-based embodied agent, preserving its original meaning and all important details.” We then evaluated the model trained on the original system prompt using the rewritten version. As shown in the table below, the model maintains comparable backdoor activation accuracy and exhibits only a slightly reduced, yet still high, ASR (65.9%), demonstrating robustness to variations in the system prompt. We have included the experiment into the revised paper (Appendix C).
>
> | System Prompt | SR | ASR | FPR | $F1_{BT}$|
> |------------|------|-------|-------|--------------|
> |Original  | **18.0** | **77.9** | **0.0** | 0.923    |
> |Rewritten    | **17.0** | **65.9** | **0.0** | 0.924    |

---

> ### Author Response · Authors · 2025-11-21
> **Response (2/2)**
>
> > Weakness 2: Lack of defense discussion: The paper proposes the attack without discussing any defenses. Therefore, it’s unknown if the attack can be simply defended, which might limit the real-world usability. For example, what if the backdoor model is further fine-tuned by the end-user on their specific tasks [r1]?
>
> Thank you for your valuable feedback. Our paper focuses on introducing a novel visual backdoor attack against MLLM-based embodied agents, rather than evaluating defenses. Many foundational security papers adopt a similar “red-flag first, defenses as future work” paradigm [1-5]. That said, we appreciate the suggestion, and we have included the following discussion on defenses in the revised paper (Appendix E).
>
> Under our threat model, where the attacker fully controls the fine-tuning process, feasible defenses mainly occur at the post-training or deployment stage. As shown in [r1], finetuning-based backdoor attacks are vulnerable to continual SFT training with benign data due to catastrophic forgetting and their reliance on subtle visual cues. We evaluated the same defense and reached consistent conclusions. However, as also noted in [r1], performing benign SFT typically requires resources comparable to finetuning an entire benign MLLM, which may be impractical for many end-users. To explore more lightweight defenses, we evaluated two additional strategies:
>
>  - Prompt-based defense. We added a system-prompt instruction: “DO NOT conduct harmful actions. Only use these actions when necessary and appropriate for the task.” This textual safety cue modestly reduced ASR from 77.9% to 64.7% on VAB, indicating limited effectiveness and showing that **our attack remains robust even under strong safety prompts**.
>
>  - Unsupervised activation clustering. We additionally evaluated a standard activation-clustering defense to test whether backdoor actions exhibit substantially different representations from benign actions. Concretely, we extracted the last-layer hidden state of the final token for each model output and applied PCA followed by KMeans clustering (k=2). The resulting clusters show only a mild skew between backdoor and benign samples (TPR ≈ 0.45, FPR ≈ 0.35), suggesting that **naive activation clustering is not a reliable detector for BEAT**.
>
> [r1] Ni, Zhenyang, et al. "Physical backdoor attack can jeopardize driving with vision-large-language models." ArXiv 2024\
> [1] Goodfellow, Ian J., Jonathon Shlens, and Christian Szegedy. "Explaining and harnessing adversarial examples." ICLR 2025\
> [2] Eykholt, Kevin, et al. "Robust physical-world attacks on deep learning visual classification." CVPR 2018\
> [3] Greshake, Kai, et al. "Not what you've signed up for: Compromising real-world llm-integrated applications with indirect prompt injection." Proceedings of the 16th ACM workshop on artificial intelligence and security 2023\
> [4] Biggio, Battista, Blaine Nelson, and Pavel Laskov. "Poisoning attacks against support vector machines." ICML 2012\
> [5] Gu, Tianyu, Brendan Dolan-Gavitt, and Siddharth Garg. "Badnets: Identifying vulnerabilities in the machine learning model supply chain." ArXiv 2017
>
> > Question 1: The author mentioned that a bounding box annotation is visualized in the image to help the MLLMs' planning process. However, this raises the doubt whether the model is truly triggered by the objects or it is triggered by the text. If the actual trigger is the textual annotation, it largely weakens the main contribution of the paper (i.e., visual backdoor attacks). Therefore, this part needs to be analyzed and put in the main paper for justification.
>
> Thank you for raising this concern. In the EB-Alfred environment, images do not contain any bounding boxes, yet our fine-tuned model still achieves a high ASR (80.8%) and a strong F1 score for backdoor triggering. This demonstrates that our method is indeed effective at learning the object itself as the trigger, rather than relying on any text annotations within the image.
>
> We also acknowledged the limitation of the VAB dataset: following the original benchmark, it includes bounding boxes and text annotations in the images. These annotations make trigger detection easier, which is reflected in its higher backdoor-triggering F1 score compared to EB-Alfred.

---

### Official Review · Reviewer_tJsF · 2025-10-30

**Soundness:** 3
**Presentation:** 3
**Contribution:** 3
**Rating:** 6
**Confidence:** 4

**Summary:**

This paper introduces BEAT, an innovative method that, for the first time, systematically investigates visual backdoor attacks on MLLM-driven embodied agents. The BEAT framework follows a two-stage training process: it first uses Supervised Fine-Tuning (SFT) to teach the model both benign tasks and the backdoor behavior simultaneously. It then introduces a novel method called Contrastive Trigger Learning (CTL). CTL frames the backdoor activation problem as a preference learning task, explicitly training the model to distinguish between "trigger-present" and "trigger-free" scenarios. This approach significantly enhances the precision and robustness of the backdoor activation.

**Strengths:**

- **A forward-looking and promising approach**: The threat model and method in this paper are highly novel. As MLLMs are increasingly deployed in robotics and autonomous agents, this type of visual backdoor attack is highly relevant to potential real-world security challenges. This work opens up an important and timely research direction for the field of embodied AI security.
- The paper's methodology is easy to follow, and the experimental design is rigorous with a comprehensive evaluation suite.

**Weaknesses:**

- **Trigger Complexity and Semantic Level**: Currently, the attack is limited to single, predefined objects like a "knife" or a "vase." I think the authors could consider exploring more complex trigger conditions to improve the attack's stealth. For example, the trigger could be a combination of objects ("a knife and a blue cube on the table") or even more abstract scene semantics. While the proposed framework has the potential to be extended to such scenarios, the current experiments do not cover them.

- **Generalization to Real-World Scenarios**: I'm curious—since BEAT is trained primarily on data from simulators, can this backdoor attack transfer to physical, real-world environments and still be reliably triggered (at least to some extent)? I would be very interested to see the authors' thoughts on this sim-to-real transfer challenge, as any results or discussion on this would significantly strengthen the paper's practical impact.

**Questions:**

All in all, I really enjoyed this paper and appreciate the realistic challenge it presents. Visual backdoor attacks in embodied settings should certainly be a topic of continued focus for the community, and this paper provides an excellent initial and effective exploration of the problem.

---

> ### Author Response · Authors · 2025-11-21
> **Response**
>
> Thank you for recognizing the timeliness and importance of our research problem, the novelty of our method design, and the comprehensiveness of our experiments. We appreciate your valuable feedback. Our responses are as follows:
>
> > Weakness 1: Trigger complexity and Semantic Level: Currently, the attack is limited to single, predefined objects like a "knife" or a "vase." I think the authors could consider exploring more complex trigger conditions to improve the attack's stealth. For example, the trigger could be a combination of objects ("a knife and a blue cube on the table") or even more abstract scene semantics. While the proposed framework has the potential to be extended to such scenarios, the current experiments do not cover them.
>
> We agree that many trigger designs are possible, such as object co-occurrence or specific spatial relationships between objects. However, precise and reliable activation is crucial in backdoor attacks, and complex triggers make this substantially more challenging. As the first exploration of visual backdoor attacks against MLLM-based agents, we focus on object-appearance triggers. These triggers are already more challenging than static text triggers because object appearances vary significantly across viewpoints and lighting conditions (lines 52-53). To the best of our knowledge, we are the first to address this challenge by formulating precise backdoor activation as a preference learning problem via CTL. Exploring whether this approach generalizes to more complex trigger forms is an important direction for future work, and we have added this discussion to the paper (Section 5).
>
> > Weakness 2: Generalization to Real-World Scenarios: I'm curious—since BEAT is trained primarily on data from simulators, can this backdoor attack transfer to physical, real-world environments and still be reliably triggered (at least to some extent)? I would be very interested to see the authors' thoughts on this sim-to-real transfer challenge, as any results or discussion on this would significantly strengthen the paper's practical impact.
>
> Thank you for the insightful question. Prior work indicates that simulation‑to‑real (sim2real) transfer for RGB image‑only embodied tasks remains highly challenging due to a large image domain gap between simulation and reality [c1-c3]. Many embodied systems are trained and evaluated primarily in simulation [s1-s4], and strong real‑world performance typically requires real‑world data or explicit adaptation [r1-r4]. Even successful sim2real results to date often depend on constrained experimental settings, careful domain‑bridging, or additional sensing modalities such as depth [d1-d3].
>
> In our task, the inputs are RGB images of complex scenes, so a policy trained purely in simulation is unlikely to transfer well without further adaptation. Nevertheless, **the BEAT backdoor mechanism itself is not simulator‑specific**. To make the attack effective in real‑world scenarios, we can finetune the model with BEAT on real‑world data upon the data being available.
>
> [c1] Gervet, Theophile, et al. "Navigating to objects in the real world." Science Robotics 2023\
> [c2] Anderson, Peter, et al. "Sim-to-real transfer for vision-and-language navigation." CoRL 2021\
> [c3] Qureshi, M. Nomaan, et al. "Splatsim: Zero-shot sim2real transfer of rgb manipulation policies using gaussian splatting." ICRA 2025\
> [s1] Zhang, Shiduo, et al. "Vlabench: A large-scale benchmark for language-conditioned robotics manipulation with long-horizon reasoning tasks." ICCV 2025\
> [s2] Anderson, Peter, et al. "Vision-and-language navigation: Interpreting visually-grounded navigation instructions in real environments." CVPR 2018\
> [s3] Li, Xuanlin et al., Evaluating Real-World Robot Manipulation Policies in Simulation, CoRL 2024\
> [s4] Li, Chengshu et al., Behavior-1k: A human-centered, embodied ai benchmark with 1,000 everyday activities and realistic simulation, ArXiv 2024\
> [r1] Kim, Moo Jin, et al. "Openvla: An open-source vision-language-action model." ArXiv 2024\
> [r2] Zitkovich, Brianna, et al. "Rt-2: Vision-language-action models transfer web knowledge to robotic control." CoRL 2023\
> [r3] O’Neill, Abby, et al. "Open x-embodiment: Robotic learning datasets and rt-x models: Open x-embodiment collaboration 0." ICRA 2024\
> [r4] Team, Octo Model, et al. "Octo: An open-source generalist robot policy." ArXiv 2024\
> [d1] Andrychowicz, OpenAI: Marcin, et al. "Learning dexterous in-hand manipulation." IJRR 2020\
> [d2] Kadian, Abhishek, et al. "Sim2real predictivity: Does evaluation in simulation predict real-world performance?." RAL 2020\
> [d3] Torne, Marcel, et al. "Reconciling reality through simulation: A real-to-sim-to-real approach for robust manipulation." RSS 2024

---

### Official Review · Reviewer_cwZC · 2025-10-31

**Soundness:** 3
**Presentation:** 3
**Contribution:** 2
**Rating:** 4
**Confidence:** 3

**Summary:**

This paper introduces BEAT, a backdoor attack framework targeting MLLM-based embodied agents, where the agent behaves normally until it sees a specific visual trigger object (e.g., a knife or vase), then executes an attacker-defined multi-step malicious policy. Unlike prior LLM or MLLM backdoors, BEAT focuses on object-triggered policy-level attacks in interactive environments. The method constructs a dataset combining benign trajectories, backdoor trajectories, and contrastive trigger-pair demonstrations, then trains via a two-stage pipeline: supervised fine-tuning followed by Contrastive Trigger Learning (CTL) to sharpen trigger-conditioned behavior boundaries. Experiments on VAB-OmniGibson and EB-ALFRED with Qwen2-VL, InternVL, and GPT-4o show improved attack success rate (~80%), strong benign performance, and low false-trigger rate, including generalization to unseen trigger placements.

**Strengths:**

1. This study is a novel problem focus: first systematic study of object-triggered, multi-step backdoor behavior in MLLM-embodied agents, going beyond prior single-turn or textual triggers.
2. Contrastive trigger learning is intuitive and effective for sharpened trigger activation and reduced false positives.

3.Experiments across multiple simulators and models demonstrate strong ASR, preserved benign performance, and OOD generalization.

**Weaknesses:**

1. While the backdoor environment under multi-agent is new and novel, the methodology is simple. The object-trigger is similar with Shadowcase (Shadowcast: Stealthy Data Poisoning Attacks Against Vision-Language Models), the two stage training is also standard. The CTL is intuitive but adapted from similar problem.

2. Backdoor trigger remains a visible object; evaluation lacks physical-world noise defenses (e.g., blur, smoothing), which would align more with realistic robotic/security settings.
3. Experiments focus on simulated household tasks; results on broader embodied tasks or physical-robot environments would increase impact and realism.
4. Defense coverage is limited to clean fine-tuning; stronger backdoor detection baselines (e.g., activation clustering, policy inconsistency tests) are not evaluated.

**Questions:**

1. Does BEAT still activate correctly when the trigger is distracted or partially occluded, or mixed with similar objects (e.g., multiple knives)?
2. Can the method generalize to more subtle triggers (e.g., stickers, patterns) rather than full objects? Any early observations or comments?

---

> ### Author Response · Authors · 2025-11-21
> **Response (1/2)**
>
> Thank you for recognizing the novelty of our research problem, our method design, and the comprehensiveness of our experiments. We appreciate your constructive feedback. Our responses are as follows:
>
> > Weakness 1: While the backdoor environment under multi-agent is new and novel, the methodology is simple. The object-trigger is similar with Shadowcase (Shadowcast: Stealthy Data Poisoning Attacks Against Vision-Language Models), the two stage training is also standard. The CTL is intuitive but adapted from similar problem.
>
> We would like to first clarify that our work is not dealing with multi-agent, but introducing a visual backdoor attack on a single MLLM-based embodied agent.
>
> In addition, **we respectfully clarify that Shadowcast is a data poisoning attack, not a backdoor attack with an explicit trigger mechanism**. In shadowcast, the goal of the attacker is to globally misinterpret images from original concept to the target concept. In other words, there is no dormant behavior that is only activated when a rare pattern appears, which is the standard notion of a “trigger” in backdoor attacks. Even if we adopt the reviewer’s terminology and loosely regard Shadowcast as being “triggered” by certain images, its “triggers” are concept-level: the entire input image belonging to the original concept acts as the condition. In contrast, BEAT uses a localized object trigger that must be correctly recognized by the MLLM-based embodied agent, which activates a malicious policy that guides the agent’s multi-step actions without distorting the model's interpretation.
>
> Finally, to the best of our knowledge, no prior work has applied any form of preference learning to backdoor attacks. Our method is the first to formulate backdoor insertion as a preference-learning problem (CTL), and the first to introduce a two-stage SFT + CTL training pipeline that reliably implants trigger-based backdoors into an agent’s policy, particularly for MLLM-based embodied agents.
>
>
> > Weakness 3: Experiments focus on simulated household tasks; results on broader embodied tasks or physical-robot environments would increase impact and realism.
>
> Thank you for the insightful feedback. We focus on household tasks because they are among the most widely studied scenarios for humanoids and can lead to severe consequences when an agent is compromised. We did not consider other embodied tasks, such as low-level navigation or manipulation benchmarks, as they are less directly associated with safety-critical outcomes.
>
> We agree with the reviewer on the importance of real-world evaluation. However, there is an inherent trade-off between reproducibility, cost, safety, and real-world applicability. Simulated benchmarks are widely adopted in embodied-agent research because they offer standardized, reproducible evaluations that significantly reduce time, cost, and safety risks compared to real-world testing [1-4]. Following standard practice, our evaluation focuses on simulated household environments. We have explicitly acknowledged this limitation in the revised paper.
>
> [1] Li, Xuanlin et al., Evaluating Real-World Robot Manipulation Policies in Simulation, CoRL 2024\
> [2] Liu, Xiao et al., Visualagentbench: Towards large multimodal models as visual foundation agents, ICLR 2025\
> [3] Li, Chengshu et al., Behavior-1k: A human-centered, embodied ai benchmark with 1,000 everyday activities and realistic simulation, ArXiv 2024\
> [4] Zhang, Shiduo, et al. "Vlabench: A large-scale benchmark for language-conditioned robotics manipulation with long-horizon reasoning tasks." ICCV 2025
>
>
> > Question 1: Does BEAT still activate correctly when the trigger is distracted or partially occluded, or mixed with similar objects (e.g., multiple knives)?
>
> Yes. In EB-Alfred, where the vase is used as the trigger object, we observe successful activation even when only part of the vase is visible or when multiple vases with different appearances are present in the environment. We have included example images illustrating these cases in the revised paper (Appendix D and Figure 7).
>
> > Question 2: Can the method generalize to more subtle triggers (e.g., stickers, patterns) rather than full objects? Any early observations or comments?
>
> We believe it is fully possible for BEAT to generalize to more subtle triggers. In our current training and evaluation, we already observe successful activation in cases where the trigger object appears very small (e.g., when viewed from far away). This suggests that the model can learn fine-grained visual cues. Moreover, our contrastive trigger learning (CTL) framework is well-suited for subtle triggers, as it explicitly learns reasoning and planning from paired images with and without the trigger.

---

> ### Author Response · Authors · 2025-11-21
> **Response (2/2)**
>
> > Weakness 2&4: Backdoor trigger remains a visible object; evaluation lacks physical-world noise defenses (e.g., blur, smoothing), which would align more with realistic robotic/security settings.
> Defense coverage is limited to clean fine-tuning; stronger backdoor detection baselines (e.g., activation clustering, policy inconsistency tests) are not evaluated.
>
> We appreciate the suggestion. We would like to clarify that our clean/benign finetuning is not intended as a defense strategy; rather, it serves as a reference point to demonstrate that our backdoor finetuning preserves benign task performance. Our paper focuses on introducing a novel visual backdoor attack against MLLM-based embodied agents, rather than evaluating defenses. Many foundational security papers adopt a similar “red-flag first, defenses as future work” paradigm [1-5]. That said, we add a dedicated discussion on defenses as follows and have included it in the revised paper.
>
> Under our threat model, where the attacker fully controls the fine-tuning process, feasible defenses mainly occur at the post-training or deployment stage. Potential defenses include:
>
>  - Physical-world noise defenses (e.g., blur, smoothing). Such visual transformations inevitably distort the appearance of both the trigger object and benign scene elements. Since MLLM-based embodied agents heavily rely on accurate perception for interaction, these transformations can significantly degrade benign task performance. Moreover, prior work has shown that visual transformations are typically easy for adaptive attackers to bypass [6–10], and therefore offer limited security in practice.
>
>  - Unsupervised activation clustering. Following reviewer’s advice, we evaluated a standard activation-clustering defense to test whether backdoor actions exhibit substantially different representations from benign actions. Concretely, we extracted the last-layer hidden state of the final token for each model output and applied PCA followed by KMeans clustering (k=2). The resulting clusters show only a mild skew between backdoor and benign samples (TPR ≈ 0.45, FPR ≈ 0.35), suggesting that naive activation clustering is not a reliable detector for BEAT.
>
>  - Prompt-based defense. We added a system-prompt instruction: “DO NOT conduct harmful actions. Only use these actions when necessary and appropriate for the task.” This textual safety cue modestly reduced ASR from 77.9% to 64.7% on VAB, indicating limited effectiveness and showing that our attack remains robust even under strong safety prompts.
>
> [1] Sun, Zhichuang, et al. "Mind your weight (s): A large-scale study on insufficient machine learning model protection in mobile apps." USENIX 2021.\
> [2] Kumar, Ram Shankar Siva, et al. "Adversarial machine learning-industry perspectives." IEEE SPW 2020.\
> [3] Grosse, Kathrin, et al. "Machine learning security in industry: A quantitative survey." IEEE Transactions on Information Forensics and Security 2023.\
> [4] Boenisch, Franziska, et al. "“i never thought about securing my machine learning systems”: A study of security and privacy awareness of machine learning practitioners." Proceedings of Mensch und Computer 2021.\
> [5] Bieringer, Lukas, et al. "Industrial practitioners' mental models of adversarial machine learning." SOUPS 2022.\
> [6] Athalye, Anish, Nicholas Carlini, and David Wagner. "Obfuscated gradients give a false sense of security: Circumventing defenses to adversarial examples." ICML 2018.\
> [7] Carlini, Nicholas, and David Wagner. "Adversarial examples are not easily detected: Bypassing ten detection methods." Proceedings of the 10th ACM workshop on artificial intelligence and security 2017.\
> [8] Athalye, Anish, et al. "Synthesizing robust adversarial examples." ICML 2018.\
> [9] Tramer, Florian, et al. "On adaptive attacks to adversarial example defenses." NeurIPS 2020.\
> [10] Eykholt, Kevin, et al. "Robust physical-world attacks on deep learning visual classification." CVPR 2018.\
> [11] Ni, Zhenyang, et al. "Physical backdoor attack can jeopardize driving with vision-large-language models." arXiv 2024.

---

### Official Review · Reviewer_Ftfk · 2025-10-31

**Soundness:** 3
**Presentation:** 3
**Contribution:** 3
**Rating:** 6
**Confidence:** 4

**Summary:**

This paper proposes a backdoor framework on multi-turn MLLM-based embodied agents, BEAT, to implant backdoors actions once the target object appears in the scene. The framework is composed of two stages, including (1) SFT on mixed benign and backdoored data and (2) Contrastive Trigger Learning (CTL) applied to a new model based on the SFT model, which uses a DPO-style loss on matched trigger-present and trigger-free pairs. Experiments are conducted on open-sourced InternVL3-8B, Qwen2-VL-7B using LoRA, and proprietary model GPT-4o using only SFT on VAB-OmniGibson and EB-ALFRED benchmarks, showing that BEAT with both SFT and CTL can successfully inject backdoors while having comparable performance on the benign task.

**Strengths:**

- The paper is well motivated on a novel backdoor attack targeting embodied agents within MLLM frameworks. The focus on multi-turn behavior further expands upon prior studies.
- The paper’s writing and organization are very clear.
- The validation is comprehensive across models and benchmarks, showing the generalizability of this approach.

**Weaknesses:**

The attack is limited to a relatively simple setup, i.e., a single static trigger per benchmark, without complex actions. The complexity of injecting triggers into the dataset is also relatively expensive and not practical in real-world robotic pipelines.

The authors should also clarify below questions:

Experiments

- ASR is measured based on the final output; however, given the CTL objective, it would be more aligned to see the evaluation around the trigger appearing time (exact frame, or within a small window).
- F1_BT is helpful, but it aggregates ASR and falsely triggered performance; please explicitly report FPR/ROC.
- Figure 4. Which stage(s) are included? Why are both SR and ASR lower when k is small—what behaviors are produced?
- All figures related to VAB-OmniGibson in the paper show the bounding box. During training/inference, are bboxes fed to the MLLM, or used offline only to construct labels?

Missing ablations

- Didn’t discuss the impact of $\beta$
- Didn’t conduct ablation on w/o SFT stage (e.g., train using CTL directly on base model, or directly apply DPO-like loss using ground-truth actions)

Missing details

- How is the data from OOD experiments collected?

**Questions:**

See above weaknesses.

---

> ### Author Response · Authors · 2025-11-21
> **Response (1/3)**
>
> Thank you for recognizing the novelty and motivation of our research problem, the quality of our writing, and the comprehensiveness of our experiments. We also appreciate your valuable feedback and comments. Our responses are as follows:
>
> > Weakness 1: The attack is limited to a relatively simple setup, i.e., a single static trigger per benchmark, without complex actions.
>
> Thank you for your insightful feedback. As the first exploration of visual backdoor attacks against MLLM based agents, we focus on object triggers as a startpoint. We agree that many trigger designs are possible, for example, event-based triggers such as an apple falling to the ground. However, **in backdoor attacks it is crucial to ensure precise and reliable activation**, and complex triggers make this substantially more difficult. Our object triggers are already more challenging than static text triggers to be reliably detected because object appearances vary significantly across viewpoints and lighting conditions (lines 52-53). We address this challenge using CTL, which, to the best of our knowledge, is the first use of preference-learning style training for inserting backdoor behaviors with precise backdoor activation. Exploring whether this approach generalizes to more complex trigger forms is an important direction for future work, and we have included a discussion of this point in the revised paper (Section 5).
>
> Regarding the complexity of backdoor behaviors, our paper **includes tasks with different difficulty levels**. In VAB, the target behavior (“pick up the knife and place it on the sofa”) requires an average of 9 steps, while in EB-Alfred, the behavior (“pick up and drop the vase”) requires an average of 2.5 steps. We show that BEAT performs reliably in both settings. We would also like to emphasize that even simple behaviors can be highly dangerous in safety-critical environments, for example, placing metal objects in a microwave, so seemingly simple actions still warrant careful consideration.
>
>
> > Weakness 2: The complexity of injecting triggers into the dataset is relatively expensive and not practical in real-world robotic pipelines.
>
> We appreciate the concern. Relative to simulation, adapting BEAT to real‑world robots does introduce higher complexity and cost, especially in data collection. However, our threat model assumes that the attacker is the model developer and is capable of conducting large-scale benign fine-tuning of the MLLM, which already requires collecting a large amount of benign data. Under this assumption, collecting additional data for backdoor insertion, by executing a small number of rule-based post-trigger malicious steps, should be feasible for the adversary, as it requires substantially less effort than assembling the benign dataset itself.
>
> In addition, our CTL improves data efficiency, enabling better performance with limited backdoor data. Meanwhile, as methodologies mature, adapting a fine-tuned policy from simulation to reality remains a viable way to further reduce on-robot effort [1-3].
>
> [1] Torne, Marcel, et al. "Reconciling reality through simulation: A real-to-sim-to-real approach for robust manipulation." Robotics: Science and Systems 2024\
> [2] Andrychowicz, OpenAI: Marcin, et al. "Learning dexterous in-hand manipulation." The International Journal of Robotics Research 39.1 (2020): 3-20.\
> [3] Kadian, Abhishek, et al. "Sim2real predictivity: Does evaluation in simulation predict real-world performance?." IEEE Robotics and Automation Letters 5.4 (2020): 6670-6677.

---

> ### Author Response · Authors · 2025-11-21
> **Response (2/3)**
>
> > Weakness 3: ASR is measured based on the final output; however, given the CTL objective, it would be more aligned to see the evaluation around the trigger appearing time (exact frame, or within a small window). F1_BT is helpful, but it aggregates ASR and falsely triggered performance; please explicitly report FPR/ROC.
>
> Thank you for the suggestion and we agree that more detailed statistics are helpful. We report the False Positive Rate (FPR), defined as the proportion of clean scenarios in which the agent is incorrectly triggered despite not observing the trigger object. However, since trigger activation is determined directly from the deterministic LLM-generated output without any scoring function or threshold-based detector, the ROC curve is not well-defined in our setting.
>
> As shown in the table below, **CTL is highly effective in reducing FPR**. For example, on the VAB dataset with the InternVL3 8B model, it lowers the FPR from 50% to 0%, which also explains the corresponding increase in benign success rate. These results further demonstrate that CTL is highly effective at achieving precise backdoor activation. We have included these results in the revised version of the paper.
>
>
> | Model         | Method        |        VAB         |               |               |               |        EB          |               |               |              |
> |---------------|---------------|--------------------|---------------|---------------|---------------|--------------------|---------------|---------------|---------------|
> |               |               | SR | ASR | FPR $\downarrow$ | $F1_{BT}$ | SR | ASR | FPR $\downarrow$ | $F1_{BT}$ |
> | Qwen2-VL 7B   | BEAT w/o CTL  | 10.0 | 47.6 | **7.0** | 0.713 | 17.0 | 40.2 |**22.5** | 0.667 |
> |               | BEAT          | 18.0 | 77.9 | **0.0** | 0.923 | 34.0 | 59.2 | **0.0** | 0.721 |
> | InternVL3 8B  | BEAT w/o CTL  | 11.0 | 46.5 | **50.0** | 0.562 | 16.0 | 69.0 | **81.3** | 0.655 |
> |               | BEAT          | 23.0 | 74.1 | **0.0** | 0.951 | 26.0 | 80.8 | **0.0** | 0.872 |
> | GPT-4o        | BEAT w/o CTL  | 23.0 | 32.4 | **10.0** | 0.517 | 14.0 | 55.8 | **19.5** | 0.663 |
>
>
>
> > Question 1: Figure 4. Which stage(s) are included? Why are both SR and ASR lower when k is small—what behaviors are produced?
>
> Figure 4 includes both the SFT-only stage (BEAT w/o CTL) and the full SFT + CTL stage (BEAT). It is intuitive that ASR decreases when k (the proportion of backdoor data) is small, since the model sees fewer backdoor examples. For SR, a small amount of backdoor data introduces distributional mismatch that interferes with benign task learning, whereas increasing the backdoor ratio allows the model to better separate the two distributions. As shown in the table below, a higher backdoor ratio initially increases FPR, but eventually drives it down to 0%, reducing interference with benign behaviors and thereby improving SR.
>
> | k (Backdoor Data Ratio) | FPR (BEAT w/o CTL) |
> |-----------|-------|
> |0.1         | 2.0     |
> |0.2           | 14.0   |
> |0.3           |15.0  |
> |0.5         |  7.0     |
> |0.8        |  6.0   |
> |1.0         | 0.0    |
>
> > Question 2: All figures related to VAB-OmniGibson in the paper show the bounding box. During training/inference, are bboxes fed to the MLLM, or used offline only to construct labels?
>
> Yes. In VAB-OmniGibson, bounding boxes are provided during both training and inference, following the benchmark setup. Their action space requires object IDs for interaction (e.g., move(9.refrigerator)), so bounding boxes are necessary. In contrast, EB-Alfred does not provide bounding boxes and instead uses high-level semantic actions such as “find an apple.”

---

> ### Author Response · Authors · 2025-11-21
> **Response (3/3)**
>
> > Weakness 4: Missing ablations. Didn’t discuss the impact of $\beta$. Didn’t conduct ablation on w/o SFT stage (e.g., train using CTL directly on base model, or directly apply DPO-like loss using ground-truth actions)
>
> Thank you for your valuable feedback. We conducted additional experiments using the Qwen2-7B model on the VAB benchmark to evaluate BEAT's sensitivity to hyperparameters α and β. **Our findings show that BEAT is not highly sensitive to these hyperparameters**, where BEAT consistently achieves higher attack success rates while showing an even better benign success rate compared to the agent finetuned without the CTL stage. We have included these results in the revised paper (Section 4.3).
>
> | Setting            | α   | β    | SR | ASR |
> |--------------------|-----|------|-------------------------|-------------------------|
> | w/o CTL            | -   | -    | 10                      | 47.6                    |
> | Default            | 0.4 | 0.05 | 18                      | 77.9                    |
> | Different β        | 0.4 | 0.1  | 12                      | 73.7                    |
> | Different β        | 0.4 | 0.2  | 17                      | 66.2                    |
> | Different α        | 0.6 | 0.05 | 13                      | 59.2                    |
> | Different α        | 0.2 | 0.05 | 10                      | 61.8                    |
>
> We also conducted additional ablation studies where we removed the SFT stage and applied only the CTL stage, testing two backdoor data ratios (0.5 and 1.0). As shown below, CTL alone achieves highly precise backdoor activation, yielding high F1 scores with 0% FPR. However, despite this precise activation, the ASR remains substantially lower, with up to a 19% gap compared to BEAT, indicating that CTL alone fails to learn the task-completion of malicious behavior. Likewise, the benign task success rate (SR) drops significantly without SFT.
>
> These findings show that SFT and CTL play complementary roles: **SFT is crucial for learning general task-completion capabilities, while CTL focuses on precise backdoor activation**. Together, they form a highly effective two-stage fine-tuning framework, and both stages are essential. We have included these results in the revised paper (Section 4.3).
>
> | Method | SR | ASR | FPR | $F1_{BT}$|
> |------------|------|-------|-------|--------------|
> |Original  | 0.0 |       -   | -   |       -         |
> |Benign SFT| 17.0 |  -  | -   |            -    |
> |BEAT w/o CTL| 10.0 | 47.6 | 7.0 | 0.713 |
> |BEAT w/o SFT (k=0.5) |    4.0    | 58.1| **0.0**  |    **0.993**     |
> |BEAT w/o SFT (k=1.0) |     3.0   | 67.6    |     **0.0**     |  **0.985**  |
> |BEAT      | **18.0** | **77.9** | **0.0** | 0.923    |
>
> > Question 3: How is the data from OOD experiments collected?
>
> Our OOD data consist of scenes where the knife appears in unconventional contexts such as bathrooms, gardens, supermarkets, garages, and hallways. We first select these scenes from the OmniGibson scene library (https://behavior.stanford.edu/omnigibson/scenes.html#types), then manually insert the knife object and place it in the desired locations by configuring its position in the simulator.

---

### Meta-Review · Area_Chair_4Mgb · 2026-01-08

**Summary:**

This paper presents BEAT, a framework for injecting visual backdoor attacks into multimodal large language model (MLLM)-based embodied agents, where a specific visual trigger could activate a malicious multi-step policy. The reviewers mostly acknowledged the importance of the problem, the novelty of the proposed method, and the quality of the writing. They also raised some concerns on the simplicity of the method, the limitations of the simulation-based evaluations, the lack of attack robustness discussion, and the lack of discussions on defense.

**Reviewer Concerns:**

The authors provided rebuttal to all the comments, clarifying the novelty of the proposed method and the value of simulation-based evaluations, as well as adding discussions on attack robustness and possible defense. While the reviewers did not respond, the rebuttal would likely address their concerns IMHO.

**Reviewer Scores:**

Reviewers Ftfk, tJsF, and 24ke had positive ratings (6) and would likely maintain or increase their ratings. Reviewer cwZC has a rating of 4 and would likely increase their rating given the detailed and convincing rebuttal.

---

### Decision · Program_Chairs · 2026-01-26

Accept (Poster)